# Neonatal Feeding Practices and SARS-CoV-2 Transmission in Neonates with Perinatal SARS-CoV-2 Exposure: A Systematic Review and Meta-Analysis

**DOI:** 10.3390/jcm14010280

**Published:** 2025-01-06

**Authors:** Kikelomo Babata, Rehena Sultana, Jean-Michel Hascoët, Riya Albert, Christina Chan, Kelly Mazzarella, Tanaz Muhamed, Kee Thai Yeo, Juin Yee Kong, Luc P. Brion

**Affiliations:** 1Division of Neonatal-Perinatal Medicine, Department of Pediatrics, University of Texas, Southwestern Medical Center, Dallas, TX 75390, USA; riya.albert@utsouthwestern.edu (R.A.); christina.chan@utsouthwestern.edu (C.C.); kelly.mazzarella@utsouthwestern.edu (K.M.); luc.brion@utsouthwestern.edu (L.P.B.); 2Centre for Quantitative Medicine, Duke-NUS Medical School, Singapore 169857, Singapore; rehena.sultana@duke-nus.edu.sg; 3DevAH, CHRU Nancy, Lorraine University, 54505 Vandoeuvre-Les-Nancy, France; j.hascoet@chru-nancy.fr; 4Thayer School of Engineering Dartmouth College, Hanover, NH 03755, USA; tanaz.k.muhamed.26@dartmouth.edu; 5Department of Neonatology, KK Women’s and Children’s Hospital, Singapore 229899, Singapore; yeo.kee.thai@singhealth.com.sg (K.T.Y.); kong.juin.yee@singhealth.com.sg (J.Y.K.)

**Keywords:** COVID-19, SARS-CoV-2, breastfeeding, breastmilk, formula, precautions, isolation, systematic review, meta-analysis

## Abstract

**Background:** The risk of neonatal SARS-CoV-2 infection from the mother’s own milk (MoM) in neonates who are exposed to maternal SARS-CoV-2 during the perinatal period remains unclear. We conducted a systematic review to assess the association between MoM feeding and neonatal SARS-CoV-2 infection in neonates who were born to SARS-CoV-2-positive pregnant persons. **Methods:** PubMed Central and Google Scholar were searched for studies published by 14 March 2024 that reported neonatal SARS-CoV-2 infection by feeding type. This search, including Scopus, was updated on 17 December 2024. The primary outcome was neonatal SARS-CoV-2 infection. The meta-analysis was conducted using a random effects model with two planned subgroup analyses: time of maternal PCR testing (at admission vs. previous 2 weeks) and dyad handling (isolation vs. some precautions vs. variable/NA). **Results:** The primary outcome was available in both arms of nine studies, including 5572 neonates who received MoM and 2215 who received no MoM. The GRADE rating was low quality, because the studies were observational (cohorts). The frequency of SARS-CoV-2 infection was similar in both arms (2.7% MoM vs. 2.2% no MoM), with a common risk ratio of 0.82 (95% confidence interval 0.44, 1.53, *p* = 0.54). No significant differences were observed in the subgroup analyses. Limitations include observational and incomplete data, other possible infection sources, small sample sizes for subgroup analyses, and neonates with more than one feeding type. **Conclusions:** Feeding MoM was not associated with an increased risk of neonatal SARS-CoV-2 infection among neonates who were born to mothers with perinatal infection. These data, along with reports showing a lack of active replicating SARS-CoV-2 virus in MoM, further support women with perinatal SARS-CoV-2 infection feeding MoM. Registration: PROSPERO ID CRD42021268576.

## 1. Introduction

More than four years have passed since the World Health Organization declared COVID-19 a global pandemic on 11 March 2020 [1,2]. The WHO reports that there have been over 760 million cases and 6.9 million deaths from SARS-CoV-2 infection since the pandemic started. This number is likely underreported [3]. This pandemic has affected everyone, including pregnant persons and newborns. While most COVID-19-infected newborns are asymptomatic or exhibit mild to moderate symptoms [4,5,6], some face critical illnesses [7,8]. However, the long-term effects of neonatal COVID-19 remain unclear.

Breastfeeding practices have been questioned since the pandemic’s onset [5,9]. Breastfeeding is a well-established protective measure against various neonatal, infant, and childhood illnesses, reducing neonatal and infant mortality and offering numerous benefits to mothers [10,11]. Since there is overwhelming evidence to support the benefits of breastfeeding for infants and mothers [10,11], any recommendations for alternatives owing to the potential risk of COVID-19 transmission would require strong justification. Nevertheless, although the risk of SARS-CoV-2 transmission to newborn infants through breastfeeding or feeding of expressed mothers’ own milk (MoM) from mothers with COVID-19 was initially unknown, concerns regarding SARS-CoV-2 transmission through breastfeeding have led to inconsistent recommendations at the beginning of the pandemic, ranging from complete separation and cessation of breastfeeding to breastfeeding with precautions (face mask). Unfortunately, early recommendations have significantly diminished the enthusiasm for breastfeeding [12]. As a result, lower breastfeeding frequencies have been reported in in association with documented maternal SARS-CoV-2 infection in several studies (including in NICUs) and in a systematic review [12,13,14,15,16]. A decrease in breastfeeding initiation has the potential to worsen infant mortality rates worldwide [17].

The current evidence suggests that SARS-CoV-2 is not transmitted through breastmilk, with most samples from infected mothers testing negative. Although some reports have detected the virus in breastmilk, there is no evidence that it is active, replicating, or contagious [18]. Vaccinated and recovered mothers demonstrate strong immune responses, producing neutralizing antibodies in their breastmilk and plasma without an increase in adverse events [19,20]. International and national guidelines have been revised accordingly and are now more supportive of breastfeeding [21,22].

In a previous systematic review, we analyzed evidence regarding delivery management interventions that are designed to prevent delivery-associated transmission of maternal SARS-CoV-2 to neonates [23]. The current systematic review aims to compare the outcomes associated with different feeding practices in neonates who were born to mothers infected with SARS-CoV-2 globally. Using a pragmatic/observational approach [24], this review has the potential to significantly improve neonatal outcomes and inform future randomized controlled trials.

## 2. Materials and Methods

We conducted this systematic review using guidelines from the Cochrane Handbook for Systematic Reviews and Preferred Reporting Items for Systematic Reviews and Meta-Analyses (PRISMA). It is registered with PROSPERO under ID CRD42021268576. The initial protocol (see Appendix A) was modified due to the evolution of evidence during the pandemic.

### 2.1. Eligibility Criteria

The inclusion criteria were randomized trials or cohort studies reporting (1) pregnant persons with a positive SARS-CoV-2 PCR test at admission or within 14 days prior to delivery, (2) information about individual neonate’s feeding type (any MoM vs. no MoM), and (3) the availability of the neonate’s SARS-CoV-2 PCR result.

### 2.2. Search STRATEGY

A detailed history of the search strategy is presented in Appendix A. We decided to only include cohorts or randomized trials to prevent selection bias associated with case reports and case series (see results using the previous strategy in the Appendix A). A search was conducted by LPB using Medline and Google Scholar for studies published from 1 January 2020 to 14 March 2024 and including neonates who were born at any GA to pregnant persons who tested positive for SARS-CoV-2 within the last 14 days of their pregnancy or immediate postpartum period. A preliminary search did not detect any relevant randomized trials. For this updated search, terms included (COVID-19 OR SARS-CoV-2) AND (neonatal OR newborn OR neonates) AND (breast milk) OR (human milk) OR breastfeeding) AND transmission AND cohort.

### 2.3. Data Extraction

Studies were selected if maternal and neonatal SARS-CoV-2 PCR testing was reported. Studies that did not report the feeding strategy for the neonate were not included. We also excluded studies where the neonate was infected after 28 days of life.

Screening and manual data extraction process were performed by LPB and KB in 2024. Information was collected using an Excel spreadsheet on demographics, timing of PCR prior to delivery, dyad handling, neonatal feeding type (MoM versus no MoM), neonatal SARS-CoV-2, and neonatal mortality. Overall, this review compared the outcome of SARS-CoV-2 infection among neonates who received any MoM versus those who did not.

### 2.4. Outcomes

The primary outcome was neonatal SARS-CoV-2 infection, confirmed by a positive RT-PCR test within 1–28 days postnatal from any sample (nasopharynx, oropharynx, saliva, stool, urine, or blood). The secondary outcome was neonatal mortality.

Neonatal mortality is reported where numbers are available. Some reported no deaths at the time of discharge or time of follow-up. This is reported in Table 1. Some cohorts made no mention of mortality; for these, we have recorded not available (NA).

### 2.5. Risk of Bias Assessment Approach

We assessed the risk of bias using the Risk of Bias in Cohort Studies by the CLARITY Group at McMaster University [38]. This tool assesses eight key domains of bias in cohort studies: (1) Cohort selection comparability, (2) Exposure assessment confidence, (3) Absence of outcome at baseline, (4) Adjustment for confounding variables, (5) Prognostic factor assessment, (6) Outcome assessment reliability, (7) Follow-up adequacy, and (8) Co-intervention similarity. K.B. and C.C. independently assessed the studies and recorded the risk of bias according to the domains listed above. Disagreements between reviewers were resolved through discussion, with LPB acting as an arbitrator. Each domain was assigned a color, with green representing a very low risk of bias, yellow a low risk of bias, orange a high risk of bias, and red a very high risk of bias. The specifics of the approach and assessment for each question are shown below in Table 2. A high risk was assigned to studies when either three or more domains were rated as very high risk or a very high-risk rating in Domain 1 (selection of exposed and non-exposed cohorts).

### 2.6. Synthesis Methods

We categorized the included interventions into two types: (1) feeding with any MoM, which included cases where the neonate received expressed MoM, was breastfed, or received formula along with breastfeeding; (2) feeding with no MoM, defined as the exclusive use of formula, donor breastmilk, or both.

The timing of maternal SARS-CoV-2 testing was classified as follows: (1) “At admission” if testing was conducted at admission, at delivery, or soon thereafter, and (2) “14 days” if testing occurred anytime within the last two weeks of pregnancy.

Dyad handling was classified as follows: (1) some precautions if a minimum of mask use was reported as the predominant approach for the cohort; (2) isolation if mothers were completely separated from their neonates; (3) variable if there were multiple predominant types of methods described; and (4) not available/unclear if the isolation or precaution methods were not explicitly described. In cohorts where a single predominant approach was utilized, the cohort was characterized by that approach, even if there were some deviations. The last 2 categories were merged for the meta-analysis.

We did not perform any data conversions. The number of mothers who breastfed and the number of neonates testing positive were obtained based on the available cohort data. This allowed us to derive relevant summary statistics, which were then used for the synthesis and meta-analysis. Cohorts with only one type of exposure (either MoM or no MoM) and outcomes with missing data and not reported by feeding type were not included in the meta-analysis. These studies are displayed in Table 1 above.

### 2.7. GRADE Assessment

The certainty of the evidence was evaluated using the GRADE framework. All included studies were cohort studies. We assessed the risk of bias, inconsistency, indirectness, imprecision, and publication bias. Sensitivity analyses were conducted to assess the impact of excluding any study with a high risk of bias on the overall findings. Inconsistency across studies was examined. Imprecision was assessed in the overall meta-analysis and in the subgroup analyses.

### 2.8. Statistical Analysis

The primary and secondary outcomes were neonatal SARS-CoV-2 infection and neonatal mortality, respectively. Both outcomes were treated as binary variables. Both outcomes were pooled using the DerSimonian–Laird random effects method. Pooled results were expressed as the common risk ratio (RR) with a 95% confidence interval (95% CI). Subgroup analyses based on the time of PCR testing in pregnancy (at admission vs. previous 2 weeks) and dyad handling in the hospital of delivery (separation, precautions, or variable/unclear) were performed for both outcomes. Continuity correction was applied for all pooled results if any included study had 0 events. The heterogeneity of the included studies was evaluated using I^2^ statistics, considering a value less than 25% as low heterogeneity, 50 to 75% as medium heterogeneity, and greater than 75% as high heterogeneity. All the analyses were performed using R (“meta” library v4.2.2) and shown in forest plots. Publication bias was assessed using a funnel plot and, if the number of studies was at least 10, Eggers’ test.

## 3. Results

### 3.1. Selection of Studies

The search conducted on 14 March 2024 yielded 1010 studies, with an additional 8 having been identified from a previous search. Among the 122 studies selected for full-text review, 13 studies met the inclusion criteria, and 9 were used for the meta-analysis, because 4 studies reported on a single feeding type only. Figure 1 provides the PRISMA flow diagram for systematic reviews, outlining the study selection process in detail. An update on 17 December 2024, expanded to include Scopus, did not yield any additional relevant studies. Four studies overlapping with Hudak [25] were excluded (Appendix A) [39,40,41,42].

### 3.2. Study/Participant Characteristics

This study included 13 cohorts from 10 countries, comprising a total of 8514 neonates (Table 1). Of these, 6299 neonates received any MoM and were tested for SARS-CoV-2 infection, with 166 (2.6%) testing positive. Of the 6299 who received any MoM, only 5572 were in studies that included both exposures (MoM and no MoM); of those, 154 (2.7% (95% CI: 2.35–3.23)) tested positive. In comparison, 2215 neonates who received no MoM were tested for SARS-CoV-2 infection, with 48 (2.2% (95% CI: 1.60–2.86)) testing positive. The risk ratio of neonatal SARS-CoV-2 infection associated with feeding MoM vs. no MoM is provided in Section 3.6.1.

### 3.3. Dyad Handling in the Hospital

There was a large array of dyad handling procedures across the studies. Various measures implemented with the intent of preventing SARS-CoV-2 transmission among neonates are described below:Full Isolation: Two studies reported the use of full isolation protocols, with mothers being completely isolated from their newborns.Some Precautions: Seven studies (four of these included in the meta-analysis) adopted varying levels of precautionary measures. These likely included the use of masks, different hygiene measures, and placement of the neonate in a bassinet or isolette 6 feet away from the mother.Variable Practices: Two studies described variable precaution/isolation practices, indicating multiple different infection control measures.Not Available or Unclear: Two studies either did not specify the type of precautionary measures or isolation practices used or did not report whether the recommended precautions were effectively implemented. Only one of those two was included in the meta-analysis.

### 3.4. Timing of PCR Testing for SARS-CoV-2 in the Pregnant Person

Of the thirteen studies, eight studies conducted SARS-CoV-2 testing on admission, at delivery, or soon thereafter, and five studies performed testing within 14 days prior to delivery (Table 1). For the nine that were included in the meta-analysis, five studies carried out testing within two weeks prior to delivery, and four tested on admission.

### 3.5. Risk of Bias Assessment

The risk of bias assessment is shown in Table 3. Eight of the included studies demonstrated an overall moderate risk of bias, while one study exhibited a high risk of bias.

### 3.6. Association of MoM with SARS-CoV-2 Infection Risk

#### 3.6.1. Title Overall Analysis

Nine studies (7787 babies) reporting both feeding types (any MoM and no MoM) were included in the meta-analysis. The forest plot (Figure 2) illustrates the effect sizes for each study and the overall effect size. The combined RR for COVID-19 infection in neonates receiving MoM compared to those receiving no MoM was 0.82 (95% CI 0.44, 1.53, *p* = 0.54). The heterogeneity among the included studies was low (Tau-squared = 0.162, I^2^ = 0%).

#### 3.6.2. Subgroup Analysis by Timing of Maternal PCR (Figure 2)

The total (random effect) RR for COVID-19 infection in neonates receiving MoM compared to those receiving no MoM was 1.15 (95% CI 0.82, 1.63) among the studies with maternal PCR within 14 days prior to delivery and 0.46 (95% CI 0.17, 1.26) among the studies with maternal PCR at admission. The test for subgroup difference was not significant (chi-square 2.87, degrees of freedom (dfs) = 1, *p* = 0.09).

#### 3.6.3. Subgroup Analysis by Dyad Handling (Figure 3)

This analysis is shown in Figure 3. Full isolation was reported in two studies: Wroblewska et al. and Shlomai [30,33]. Both did not show any significant differences in the rate of infection between infants who were fed any MoM and those who were fed no MoM.

The total (random effect) RR for COVID-19 infection in neonates receiving MoM compared to those receiving no MoM was 0.84 (95% CI 0.42, 1.67) among studies with variable/NA practice; 0.46 (95% CI 0.04, 4.73) among studies with some precautions; and 1.00 (95% CI 0.03, 26.63) among studies without precautions. The test of the between-subgroup difference was not significant (chi-square 0.26, dfs = 2, *p* = 0.88). Because of the small sample size, there was a large CI for most of the observations; therefore, we are very unsure of the results.

**Figure 3 jcm-14-00280-f003:**
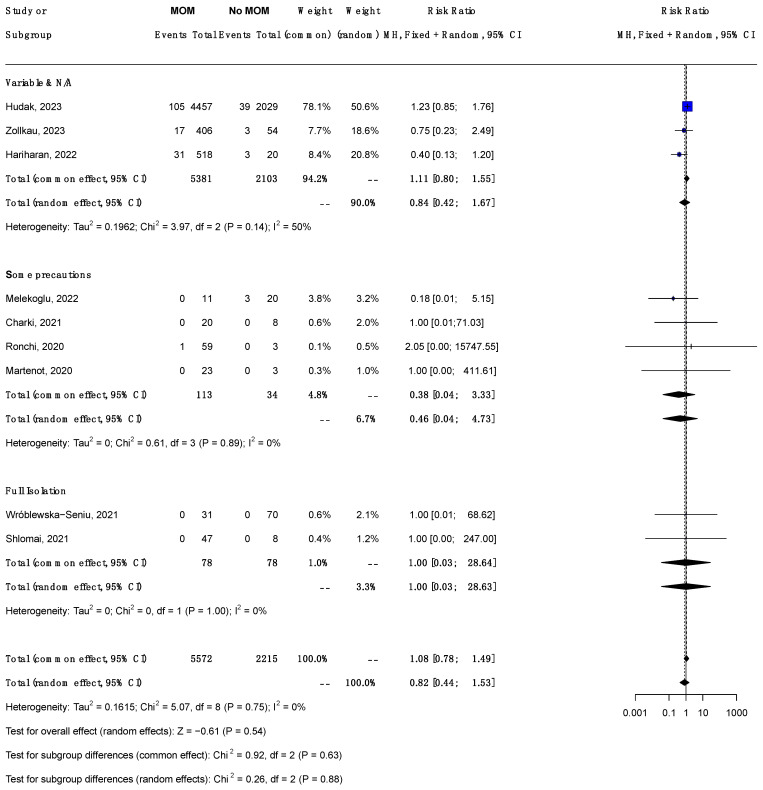
Forest plot showing the risk ratio of neonatal COVID-19 infection (confirmed by PCR) in neonates receiving any MoM vs. those receiving no MoM; subgroup analysis by dyad handling. Abbreviations: MH, Mantel–Haenszel; CI, confidence interval; dfs, degrees of freedom.(Studies included [25,26,28,29,30,32,33,34,35]). Foot note: **Variable practices** refer to different infection control measures, while **NA (Not Available or Unclear)** indicates instances where data on precautions or isolation protocols were either unspecified or not reported. **Full isolation** involved complete maternal separation from newborns, **Some precautions** refer to varying precautionary measures. **Common Effect** refers to a statistical model assuming a single true effect size across all studies, while **Random Effect** assumes the true effect size may vary between studies due to heterogeneity.

### 3.7. Sensitivity Analysis

The sensitivity analysis that was carried out excluding the study with a very high risk of bias [26] did not show a change in the overall results. The common RR including all studies was 0.82 (95% CI 0.44, 1.53), compared with 0.79 (95% CI 0.44, 1.76) after excluding the single study with a high risk of bias.

### 3.8. Publication Bias

We used 9 out of 13 studies for the meta-analysis, as 4 of the studies included only one arm. Since these four studies included 727 neonates and our meta-analysis included 7787 neonates, this was unlikely to affect our results. The funnel plot (Figure 4) indicates that the studies are relatively well distributed around the pooled effect size, suggesting no strong evidence of publication bias. We could not perform Eggers’ test, because there were less than 10 studies.

### 3.9. GRADE Rating

The overall GRADE rating was low quality of evidence, largely due to the observational nature of the included studies. All studies included in the meta-analysis have some limitations, particularly concerning the risk of bias, which was rated as high in one study and moderate in all others. The sensitivity analysis excluding the high-risk study showed no significant change in outcomes, which supports the robustness of the findings despite the inherent biases.

The evidence did not show any inconsistency or indirectness, and there was no indication of publication bias. However, imprecision was evident, as indicated by the large CI, especially in subgroup analyses where sample sizes were smaller.

In summary (Table 4), the certainty of the evidence presented in this systematic review is low, primarily because no randomized trial is available.

The certainty of the preliminary results presented in Appendix A was very low, because these only included case reports and case series. These data did not show any evidence of an association of feeding MoM or setting of feeding with neonatal SARS-CoV-2 infection.

## 4. Discussion

Our study aimed to explore the association between neonatal feeding types—specifically, MoM versus no MoM—and the risk of SARS-CoV-2 infection in neonates who were born to mothers with confirmed SARS-CoV-2 infection within the last 2 weeks of pregnancy. We found no significant difference in the frequency of neonatal SARS-CoV-2 infection between those who were fed MoM and those who were not fed MoM. Additionally, the timing of infection (at admission versus the 14 days prior to delivery) and type of dyad handling (isolation, some precautions, or variable/NA) were not associated with an increased risk of neonatal infection. These findings contribute important insights to guide breastfeeding recommendations during the COVID-19 pandemic, highlighting the safety of MoM even in the context of maternal SARS-CoV-2 infection.

The observational nature of our study is a limitation, since no randomized controlled trial was available. Instead, we employed a pragmatic study design, similar to that used by the Vermont Oxford Network [24]. This allowed us to incorporate data from multiple international cohorts and thus enhance the generalizability of our findings. The multinational nature of our cohort, which includes neonates from 10 different countries, lends additional strength to our conclusions.

Our findings align with a previous systematic review examining the safety of breastfeeding during maternal SARS-CoV-2 infection. In a systematic review of 176 cases conducted in 2020, Raschetti et al. reported that no significant association was found between breastfeeding and a risk of neonatal SARS-CoV-2 infection [44]. However, they reported that a lack of mother–neonate separation from birth was associated with increased odds of late infection (adjusted odds ratio 6.6, 95% CI 2.6–16, *p* < 0.0001) [44]. These latter findings contrast with the results in our systematic review of cohort studies, in which we found no association between dyad handling and risk of neonatal infection. Similarly, Morniroli et al. [45] reported in a systematic review of prospective observational studies that the rate of mother-to-neonate transmission was similar in those with (1.4%, CI 0.8–2%) and those without rooming-in (1.3%, CI 0.0–2.7%) and was not significantly different in those utilizing two or more protective measures (1.0%, CI 0.3–1.7%) versus one or no measures (3.2%, CI 1.2–5.2%). Another systematic review of breastfeeding with maternal SARS-CoV-2 infection was conducted by Boukara et al. This study supported breastfeeding, skin-to-skin contact, and rooming-in with diligent hand hygiene, wearing masks, and cleansing of breasts only when necessary [46]. These results support the benefit of maintaining breastfeeding practices with appropriate infection control measures.

Concerns about the safety of breastfeeding in SARS-CoV-2-positive mothers were raised by isolated reports of viral RNA detection in breastmilk. However, studies evaluating the presence of SARS-CoV-2 in breastmilk have largely shown reassuring results. For instance, Kunjumon et al. [41] reported that 18 of 19 breastmilk samples from SARS-CoV-2-positive mothers were negative, and even in the single case where viral RNA was detected, the neonate remained uninfected. A similar pattern was observed in a living systematic review of 37 articles, where 43 of 46 milk samples were negative, and no infectious viral particles were found in any positive samples [47]. Powell [18] and Of Pang [48] also emphasized that while viral RNA has occasionally been detected, infectious particles have not been isolated, suggesting a limited risk of transmission through breastfeeding. Additional evidence supporting the safety of MoM comes from Krogstad et al. [49], who analyzed breastmilk samples from 110 lactating women and found that while SARS-CoV-2 RNA was present in some samples, no infectious viruses or sub-genomic RNA was detected. Thus, so far, no study has shown the presence of infectious COVID-19 viral particles in breastmilk. Moreover, breastmilk contains neutralizing antibodies against SARS-CoV-2, derived both from prior infection and vaccination, which may confer additional protective benefits to neonates [50,51,52,53].

### Strengths and Limitations

The strengths of our systematic review are our methods following the Cochrane and PRISMA guidelines; the inclusion of cohort studies only, thereby limiting the selection bias associated with cases and case series; the limitation to studies using PCR for maternal and neonatal COVID-19 infection rather than antigen testing or symptoms only, thereby limiting false positives; the meta-analysis limited to cohorts with two arms (MoM vs. no MoM), thereby allowing for the calculation of the combined RR; the large sample size for the primary outcome, thereby reducing imprecision; the multinational scope without language exclusion, thereby enhancing the generalizability of our findings; and the sensitivity analysis, thereby showing the robustness of the results. The inclusion of various dyad handling types and feeding exposures allowed us to better characterize the role of different factors in neonatal SARS-CoV-2 transmission.

Several limitations need to be acknowledged. The certainty of the evidence presented in this systematic review was low, based on the GRADE assessment. Only observational studies were included, thereby limiting the analysis to association and not causation. All included cohort studies had at least a moderate risk of bias and potential residual confounding. Cohorts included in this systematic review lacked the granularity observed in case reports or case series but had no evidence of selection bias. While the cohort design allowed for a pragmatic evaluation of the research question, the inclusion of only 9 of the 13 available studies in the meta-analysis limits the comprehensiveness of the findings. Furthermore, the small sample sizes in the subgroup analyses led to imprecision, reflected in the wide CI, particularly in dyad handling subgroups. Additionally, we could not ascertain the specific source of neonatal infection, as droplet transmission remains the predominant route of SARS-CoV-2 spread. In addition, our study design did not allow us to compare successive viral subtypes during the pandemic. Lastly, our study period included varying phases of vaccine availability, from none to increasing uptake, which could have influenced the infection risk but was not accounted for in our analysis [54,55]. The certainty of the evidence of the preliminary results based on case reports and case series was very low.

We analyzed dyad handling as reported in the cohort studies but acknowledge that specific infection control measures may have varied significantly at the individual level. This heterogeneity in protocols and their application limits our ability to draw firm conclusions about the most effective infection control strategies. Future research should aim for standardized reporting and consistency in intervention protocols to better inform best practices.

## 5. Conclusions

Our findings on feeding MoM contribute to the growing body of evidence supporting breastfeeding during maternal SARS-CoV-2 infection. Given the low certainty of evidence, additional well-designed studies are warranted to confirm these findings and further elucidate the role of vaccination in mitigating neonatal infection risk. Future research should also focus on refining infection control practices during breastfeeding to optimize the outcomes for both mother and infant.

## Figures and Tables

**Figure 1 jcm-14-00280-f001:**
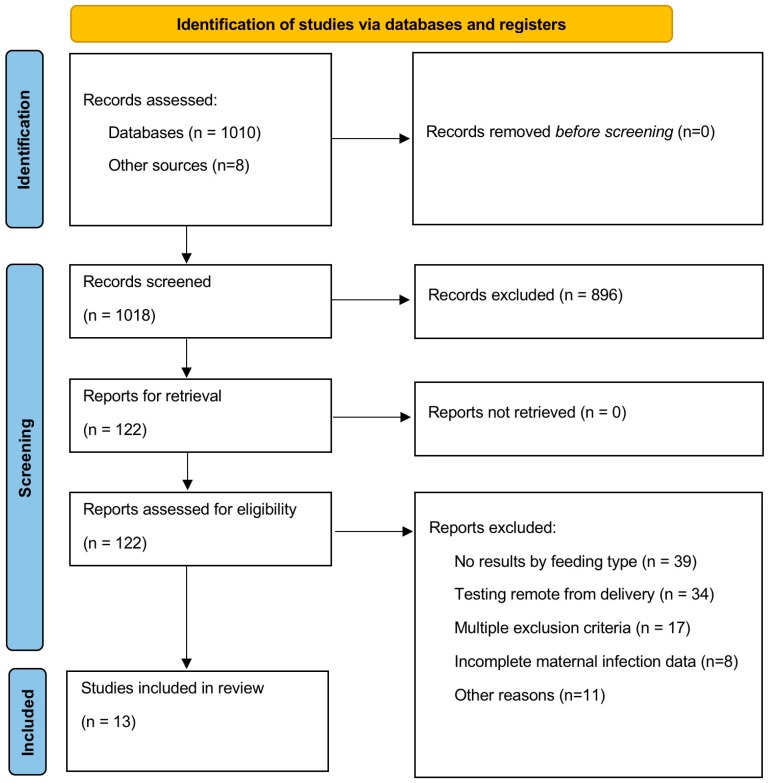
PRISMA [43] Flow diagram.

**Figure 2 jcm-14-00280-f002:**
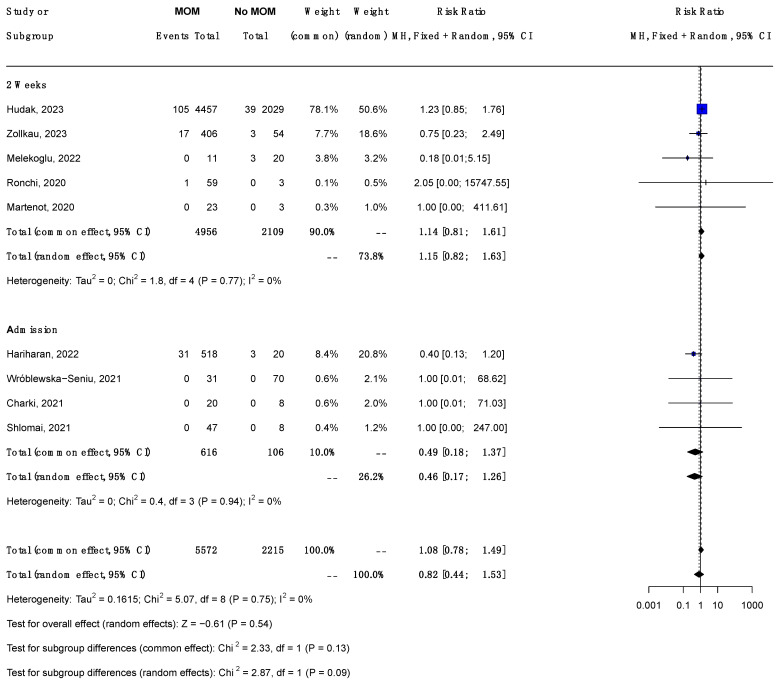
Forest plot showing the risk ratio of neonatal COVID-19 infection (confirmed by PCR) in neonates receiving any MoM vs. those receiving no MoM; subgroup analysis by timing of maternal PCR testing. Abbreviations: MH, Mantel–Haenszel; CI, confidence interval; dfs, degrees of freedom. (Studies included [25,26,28,29,30,32,33,34,35]), Foot note: MOM: Mother’s Own Milk. The timing of maternal SARS-CoV-2 testing was classified as follows: **Admission** refers to testing conducted at the time of admission, at delivery, or immediately thereafter, and **2 weeks** indicates testing performed anytime within the last two weeks of pregnancy. **Common Effect** refers to a statistical model assuming a single true effect size across all studies, whereas **Random Effect** assumes that the true effect size may vary between studies due to heterogeneity.

**Figure 4 jcm-14-00280-f004:**
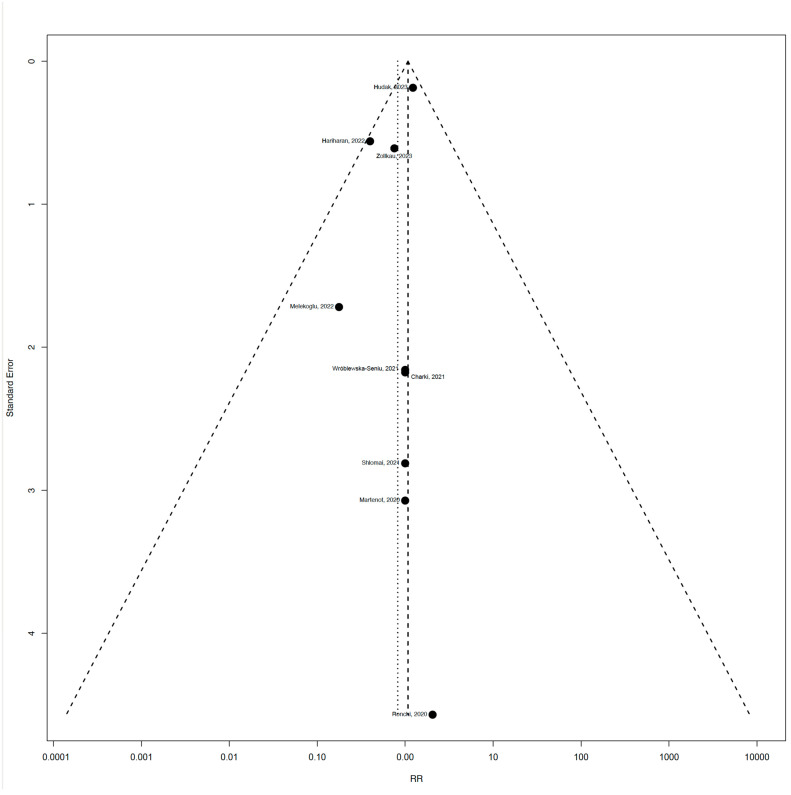
Funnel plot showing the standard error of the RR on the y-axis against the RR on the *x*-axis for studies included in the meta-analysis. Abbreviation: RR, risk ratio. (Studies included [25,26,28,29,30,32,33,34,35]).

**Table 1 jcm-14-00280-t001:** Description of included studies.

Author, Study Year, Country	Subjects	Design	Dyad Handling	Maternal SARS-CoV-2 Test Timing	Neonatal Mortality (N)	Mothers (N)	Neonates (N)	SARS-CoV-2-Positive Neonates (MoM) n/N (%)	SARS-CoV-2-Positive Neonates (No MoM) n/N (%)
Hudak [25] (2023, US)	Term and Preterm	National cohort study	Variable	14 days	29 *	7524	6236	105/4457 (2.4)	39/2029 (1.9)
Zollkau [26] (2023, Austria & Germany)	Term and Preterm	Prospective cohort study	Variable	14 days	7 *	842	460	17/406 (4.2)	3/54 (5.6)
^&^ Singh [27] (2023, India)	Term and Preterm	Cohort study	Some precautions	Admission	None reported	396	394	0/394 (0)	-
Melekoglu [28] (2022, Turkey)	Term and Preterm	Retrospective cohort study	Some precautions	14 days	None reported	30	31	0/11 (0)	3/20 (15)
Hariharan [29] (2022, India)	Term	Retrospective cohort study	NA	Admission	7 ^~^	556	531	31/518 (6.0)	3/20 (15)
Wróblewska-Seniu [30] (2021, Poland)	Term and Preterm	Retrospective cohort study	Isolation	Admission	None reported ^	101	101	0/31 (0)	0/70 (0)
^&^ Ferreira [31] (2021, Portugal)	Term and Preterm	Retrospective cohort study	Some precautions	Admission	1 *	79	81	0/81 (0)	-
Charki [32] (2021, India)	Term and Preterm	Prospective observational study	Some precautions	Admission	None reported	26	28	0/20 (0)	0/8 (0)
Shlomai [33] (2021, Israel)	Term	Retrospective cohort study	Isolation	Admission	1 *	55	55	0/47 (0)	0/8 (0)
Ronchi [34] (2020, Italy)	Term and Preterm	Prospective multi-center cohort	Some precautions	14 days	None reported	61	62	1/59 (1.7)	0/3 (0)
Martenot [35] (2020, France)	Term	Retrospective study	Some precautions	14 days	None reported	26	26	0/23 (0)	0/3 (0)
^&^ Kalamdani [36] (2020, India)	Term	Retrospective cohort study	NA	Admission	None reported	185	185	12/185 (6.5)	-
^&^ Khan [37] (2020, Pakistan)	NA	Retrospective cohort study	Some precautions	Admission	None reported	66	67	0/67 (0)	-
Overall Total 8514								166/6299 (2.6)	48/2215 (2.2)
^#^ Meta-analysis Total 7787								154/5572 (2.7)	48/2215 (2.2)

*, None that were COVID-19-related; ^, at postdischarge follow-up; -, all babies in this group received MoM only; NA, not available; ^#^, for 9 of the 13 studies that had feeding reported in both arms; ^&^, not included in meta-analysis; ^~^, no additional information provided.

**Table 2 jcm-14-00280-t002:** Risk of bias approach.

Risk of Bias Domain	Question	Approach/Assessment
1. Selection of exposed and non-exposed cohorts	Were cohorts drawn from the same population?	Green: Similar care and patients. Yellow–Orange: Same patients, but not all information on care is available. Red: Different care/patient populations.
2. Assessment of exposure	Can we be confident in the assessment of exposure?	Green: PCR—positive pregnant person (included in eligibility criteria).
3. Outcome of interest at study start	Can we be confident that the outcome of interest was not present at the start of the study?	Green: Outcome assessed at birth or within <24 h and repeated after 24 h. Yellow: Outcome assessed at any other time.
4. Matching of cohorts for variables	Did the study match exposed and unexposed cohorts for all variables associated with the outcome?	Green: Matching performed. Red: No matching performed.
5. Prognostic factors	Can we be confident in the assessment of prognostic factors?	Green: Data from validated databases or thorough individual assessments. Yellow: Data collected either prospectively or retrospectively with central queries. Orange: Data collected retrospectively without queries.
6. Outcome assessment	Can we be confident in the assessment of outcomes?	Green: PCR testing was carried out and reported for neonates by exposure.
7. Follow-up of cohorts	Was the follow-up of cohorts adequate?	Green: 100% follow-up. Yellow: 80–99% follow-up. Orange: 50–79% follow-up. Red: Less than 50% follow-up.
8. Co-interventions	Were co-interventions similar between groups?	Green: Co-interventions were similar. Yellow–Orange: Minor differences not fully documented. Red: Few or no relevant co-interventions that might influence the outcome of interest are documented to be similar in the exposed and unexposed groups.

**Table 3 jcm-14-00280-t003:** Risk of bias assessment.

Author, Year	Question 1	Question 2	Question 3	Question 4	Question 5	Question 6	Question 7	Question 8	Overall Risk of Bias
Hudak, 2023 [25]	🟡	🟢	🟡	🔴	🟡	🟢	🟡	🔴	Moderate risk
Zolkau, 2023 [26]	🔴	🟢	🟡	🔴	🔴	🟢	🔴	🔴	High risk
Hariharan, 2022 [29]	🟢	🟢	🟡	🔴	🔴	🟢	🟡	🟠	Moderate risk
Melekoglu, 2022 [28]	🟢	🟢	🟡	🔴	🔴	🟢	🟢	🟢	Moderate risk
Charki, 2021 [32]	🟢	🟢	🟡	🔴	🔴	🟢	🟢	🟢	Moderate risk
Ronchi, 2020 [34]	🟢	🟢	🟢	🔴	🔴	🟢	🟢	🟢	Moderate risk
Martenot, 2020 [35]	🟢	🟢	🟡	🔴	🔴	🟢	🟢	🟢	Moderate risk
Wroblewska, 2021 [30]	🟢	🟢	🟡	🔴	🔴	🟢	🟢	🟢	Moderate risk
Shlomai, 2021 [33]	🟢	🟢	🟡	🔴	🔴	🟢	🟢	🟢	Moderate risk

Legend: 🟢, very low risk of bias; 🟡, low risk of bias; 🟠, high risk of bias; 🔴, very high risk of bias. Question 1: Was Selection of Exposed and Non-Exposed Cohorts Drawn from the Same Population? Question 2: Can We be Confident in the Assessment of Exposure? Question 3: Can We Be Confident that the Outcome of Interest Was Not Present at Start of Study? Question 4: Did the Study Match Exposed and Unexposed for All Variables That Are Associated with the Outcome of Interest or Did the Statistical Analysis Adjust for These Prognostic Variables? Question 5: Can We Be Confident in the Assessment of the Presence or Absence of Prognostic Factors? Question 6: Can We Be Confident in the Assessment of Outcome? Question 7: Was the Follow-Up of Cohorts Adequate? Question 8: Were co-Interventions Similar Between Groups?

**Table 4 jcm-14-00280-t004:** Summary of findings.

Rate of SARS-CoV-2 Infection in No MoM Compared to Any MoM					
Outcome	Study Population	No MoM	Any MoM	Relative Effect (95% CI)	Certainty of the Evidence (GRADE)
Neonatal SARS-CoV-2 infection	7787	2215	5572	0.82 (0.44–1.53)	Low certainty
Patients or population: Neonates born to SARS-CoV-2-positive Mothers Intervention: Any MoM Comparison: No MoM					
GRADE Domains Low quality of evidence, as these are cohort studies. Risk of bias: High in one study, moderate in all other studies. Sensitivity analysis: No change in results after removal of high-risk study. Inconsistency: No evidence of this. Indirectness: None. Imprecision: Large CI despite large sample size. Especially large for subgroup analysis by type of dyad handling, due to small sample size. Publication bias: No evidence.					

Abbreviations: MoM, mother’s own milk; CI, confidence interval.

## Data Availability

The templates for data collection and datasets generated during and/or analyzed during the current study are available from the corresponding author on reasonable request. Data are also extractable from the references list in Table 1.

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
