# Peer review of "Neonatal Feeding Practices and SARS-CoV-2 Transmission in Neonates with Perinatal SARS-CoV-2 Exposure: A Systematic Review and Meta-Analysis"

_jcm, 2025, doi:10.3390/jcm14010280_

Round 1

Reviewer 1 Report

Comments and Suggestions for Authors

Specific Comments:

1. Line 30: There is an unnecessary gap between lines 30 and 32. Please address this formatting issue.

2. Line 40: Enrich the keywords section by including terms such as systematic review and meta-analysis.

3. Lines 64–65: Support the claim, “As a result, lower breastfeeding frequencies have been reported since the pandemic began,” with scientific literature. The currently cited reference does not provide sufficient evidence or reasoning to substantiate this claim.

4. Lines 92–93: Note that many guidelines recommend searching at least three databases to ensure the comprehensiveness of a systematic review or meta-analysis. 

5. Lines 96–98: Revise the manuscript to align with the latest PRISMA guidelines. For instance, include the full search algorithms used for each database in the supplementary materials or methods section.

6. Line 103: The abstract mentions that the search period is up to March 14, 2024, which may now be outdated. Please reevaluate the literature and update the search to include the most recent studies.

7. Line 263: Provide confidence intervals (CIs) for the reported I² values.

Author Response

  1. Line 30: There is an unnecessary gap between lines 30 and 32. Please address this formatting issue.

A: This was corrected.

  1. Line 40: Enrich the keywords section by including terms such as systematic review and meta-analysis.

A: We added systematic review and meta-analysis as suggested.

  1. Lines 64–65: Support the claim, “As a result, lower breastfeeding frequencies have been reported since the pandemic began,” with scientific literature. The currently cited reference does not provide sufficient evidence or reasoning to substantiate this claim. This is now located on lines 69-71.

A: We added several references to substantiate this claim.

  • Preszler J, Schriever M, Terveen M. Effects of the COVID-19 Pandemic on Breastfeeding Rates in a Single Tertiary Health Center. S D Med 2022;75(6):263–267
  • Fucile S, Heath J, Dow K. Impact of the Covid-19 Pandemic on Breastfeeding Establishment in Preterm Infants: An Exploratory Study. Neonatal Netw 2023;42(1):7–12
  • Ruiz MT, Oliveira KF, Azevedo NF, Paschoini MC, Rodrigues WF, Oliveira CJ, Oliveira JF, Fonseca LM, Wernet M. Breastfeeding prevalence in newborns of mothers with COVID-19: a systematic review. Revista Brasileira de Enfermagem. 2023 Jul 31;76(Suppl 1):e20220173.
  • Marín Gabriel, M., Martín Lozoya, S., de las Heras Ibarra, S. et al. Association of the presence of a COVID-19 infection at the time of birth and the rates of exclusive breastfeeding upon discharge in BFHI hospitals: a multicenter, prospective cohort study. Int Breastfeed J 18, 54 (2023). https://doi.org/10.1186/s13006-023-00590-0
  • Shah PS, Joynt C, Håkansson S, Narvey M, Navér L, Söderling J, Yang J, Beltempo M, Stephansson O, Fell DB, Money D, Ting JY, Norman M; for CNN. Infants Born to Mothers Who Were SARS-CoV-2 Positive during Pregnancy and Admitted to Neonatal Intensive Care Unit. Neonatology. 2022;119(5):619-628. doi: 10.1159/000526313. Epub 2022 Sep 9. PMID: 36088904; PMCID: PMC9747725.

  1. Lines 92–93: Note that many guidelines recommend searching at least three databases to ensure the comprehensiveness of a systematic review or meta-analysis. 

A: We added a search of Scopus database done on 12/17/2024; this yielded no additional relevant cohort. This is mentioned now in line 25 in the abstract and in line 205 under the selection of studies section.

  1. Lines 96–98: Revise the manuscript to align with the latest PRISMA guidelines. For instance, include the full search algorithms used for each database in the supplementary materials or methods section.

A: We added to the Supplement a detailed list of search algorithms used in this systematic review (Supplement 2). The latest keywords used for updating the search are as follows: (covid-19 OR SARS-cov-2) AND (neonatal OR newborn OR neonates) AND ((breast milk) OR (human milk) OR breastfeeding) AND transmission AND (cohort OR cohorts or randomized clinical trial)

  1. Line 103: The abstract mentions that the search period is up to March 14, 2024, which may now be outdated. Please reevaluate the literature and update the search to include the most recent studies.

A: The PubMed Central and the Google Scholar searches were updated on 12-17-2024. No additional cohort was found. An additional search using Scopus conducted on 12-17-2024 yielded 1392 references but no additional relevant study. The abstract (lines 25-26) and selection of studies (lines 205-206 )have been updated to reflect this.

  1. Line 263: Provide confidence intervals (CIs) for the reported I² values.

A: Our statistician is on sabbatical until January. We will share the confidence intervals from her as soon as we receive them.

We thank Reviewer 1 for her/his helpful comments on our manuscript.

Reviewer 2 Report

Comments and Suggestions for Authors

Dear Authors:

The topic of this work is highly relevant and important for informing future strategies related to neonatal feeding in critical situations such as the COVID-19 pandemic.

The study aimed to analyse the relationship between breastfeeding and neonatal SARS-CoV-2 infection in newborns of mothers who tested positive for the virus. Based on the bibliographic review conducted, it is observed that only two articles were included, both reporting conditions of complete maternal isolation during the study. Consequently, the findings of this meta-analysis primarily only can reflect the controlled conditions of mother-infant dyad handling rather than providing robust evidence about the inherent safety of breast milk or supporting a clear decision on whether or not to feed newborns with breast milk.

In this context, I believe the principal conclusion presented in the abstract, "Breastfeeding with breast milk was not associated with an increased risk of neonatal SARS-CoV-2 infection among newborns of mothers with perinatal infection. These data, along with reports showing the absence of active replicating SARS-CoV-2 virus in breast milk, further support breastfeeding," is somewhat "overstated."

At the first approach you only consider two categories (line 149): “We categorized the included interventions into two types: (1) Feeding any MoM: these included cases where the neonate received expressed MoM, was breastfed, or received formula along with breastfeeding; (2) Feeding no MoM, defined as the exclusive use of formula, donor breastmilk or both.” I consider that the most important aspect for this analysis would be verifying possible differences between the two subgroups within your first category (cases where the neonate received expressed MoM vs. those that were breastfed or received formula along with breastfeeding, or with some contact with the mother).

Only when you analyse the subgroup (line 282): “Subgroup analysis by dyad handling: Full isolation included two studies: Wroblewska et al. and Shlomai [26,29]. do you have the opportunity to verify this distinction. But the results obtained with this two articles suggest that despite full isolation, neonatal infection is observed in mom and any MoM ? “Both did not show any significant differences in rate of infection by any MoM vs. no MoM,”

As you rightly pointed out, there are not enough randomized studies to adequately address this critical question. Considering that your study focuses on the outcome of whether the infant becomes infected and its association with breastfeeding, it lacks the granularity needed to address the core issue: under what specific conditions is breastfeeding provided?

From the perspective of conducting the systematic review of the literature, the methodology is very clear, well-detailed, and even registered in PROSPERO. I have no criticism regarding this aspect of the study. However, while your work gathers a substantial amount of information, it does not, in fact, provide truly novel insights to guide decisions about whether to breastfeed or not and under what specific conditions it should be done.

You refer  a previous systematic review (lane 74) were you analyzed evidence about delivery management interventions designed to prevent delivery-associated transmission of maternal SARS-CoV-2 to neonates [Ref.19]. Even considering that Breastfeeding and Delivery management are distinct yet interconnected components of neonatal care, especially in the context of preventing viral transmission, the outcome that you analysed depends on both, the discussion or clinical recommendations, integrating them under a unified framework, such as "maternal-neonatal infection prevention strategies", provides a more holistic view. It is understandable that your findings may be limited by the available articles, which often address breastfeeding and delivery management separately. However, there are likely studies that consider both aspects together. Do you consider that this separation might have limited your ability to fully capture potential influences on neonatal infection rates? Could delivery management practices have impacted breastfeeding conditions, thereby affecting your results?

Please consider further developing and refining the discussion, conclusion, and future perspectives, including proposed approaches for new studies.

Kind Regards

Author Response

Reviewer 2

Dear Authors:

The topic of this work is highly relevant and important for informing future strategies related to neonatal feeding in critical situations such as the COVID-19 pandemic.

The study aimed to analyse the relationship between breastfeeding and neonatal SARS-CoV-2 infection in newborns of mothers who tested positive for the virus. Based on the bibliographic review conducted, it is observed that only two articles were included, both reporting conditions of complete maternal isolation during the study. Consequently, the findings of this meta-analysis primarily only can reflect the controlled conditions of mother-infant dyad handling rather than providing robust evidence about the inherent safety of breast milk or supporting a clear decision on whether or not to feed newborns with breast milk.

A: The reviewer states that the goal of this systematic review was to assess whether to support breastfeeding with SARS-CoV-2 infection. We respectfully disagree with the reviewer. This was not our goal. The specific conditions of breastfeeding with COVID-19 were assessed in another systematic review (Boukoura et al, ref 41), which included case series and cohorts. In contrast, as stated lines 80-84, we aimed “to compare the outcomes associated with different feeding practices in neonates born to mothers infected with SARS-CoV-2 globally”. Thus, we analyzed whether feeding mother’s own milk (MoM) was associated with transmission of the virus to the neonate. Feeding MoM is not the same as breastfeeding.

In this context, I believe the principal conclusion presented in the abstract, "Breastfeeding with breast milk was not associated with an increased risk of neonatal SARS-CoV-2 infection among newborns of mothers with perinatal infection. These data, along with reports showing the absence of active replicating SARS-CoV-2 virus in breast milk, further support breastfeeding," is somewhat "overstated."

A; For the sake of clarity, we changed the last words of the abstract into “women with perinatal SARS-CoV-2 infection feeding MoM”(line 39).

At the first approach you only consider two categories (line 149): “We categorized the included interventions into two types: (1) Feeding any MoM: these included cases where the neonate received expressed MoM, was breastfed, or received formula along with breastfeeding; (2) Feeding no MoM, defined as the exclusive use of formula, donor breastmilk or both.” I consider that the most important aspect for this analysis would be verifying possible differences between the two subgroups within your first category (cases where the neonate received expressed MoM vs. those that were breastfed or received formula along with breastfeeding, or with some contact with the mother).

A; This level of granularity is not available at individual level in the cohort studies available in this review. For most studies, feeding MoM was only a secondary, not a primary variable. However, such granularity was available in case reports and case series presented in preliminary data, Supplement 2, pages 13-14.

Only when you analyse the subgroup (line 282): “Subgroup analysis by dyad handling: Full isolation included two studies: Wroblewska et al. and Shlomai [26,29]. do you have the opportunity to verify this distinction. But the results obtained with this two articles suggest that despite full isolation, neonatal infection is observed in mom and any MoM ? “Both did not show any significant differences in rate of infection by any MoM vs. no MoM,”

A; We have verified this distinction. We changed the text as follows on lines 294-296: “Both did not show any significant differences in rate of infection between infants fed any MoM and those fed no MoM”. This point was expressed in the Strengths and Limitations section of the manuscript (lines 426-428). “We could not ascertain the specific source of neonatal infection, as droplet transmission remains the predominant route of SARS-CoV-2 spread.”

As you rightly pointed out, there are not enough randomized studies to adequately address this critical question. Considering that your study focuses on the outcome of whether the infant becomes infected and its association with breastfeeding, it lacks the granularity needed to address the core issue: under what specific conditions is breastfeeding provided?

A; Granularity needed to definitely answer the question you raise is not available in cohorts. No randomized trials are available to answer the question you are raising: “Is breastfeeding safe?” This question was addressed in another systematic review published by Boukara et al, which we included in the discussion.

The question we are asking is: “Is feeding MOM safe?”

In preliminary data included in the revised supplement, granularity was available in case series and case reports (Pages 13-14, Tables 1 and 2). These data suggested there was no evidence of association of neonatal SARS-CoV-2 infection with either type or setting of feeding.

However, the high percentage of neonatal SARS-CoV-2 infection compared with the largest available cohort (Hudak et al 2023, ref 25) suggested selection bias of case reports. This is why we decided to only include cohorts in the final version of the systematic review presented in the main document.  We mention these and other limitations in the abstract and the discussion on lines 417-419: “Cohorts included in this systematic review lacked granularity observed in case reports or case series but had no evidence for selection bias”.

From the perspective of conducting the systematic review of the literature, the methodology is very clear, well-detailed, and even registered in PROSPERO. I have no criticism regarding this aspect of the study. However, while your work gathers a substantial amount of information, it does not, in fact, provide truly novel insights to guide decisions about whether to breastfeed or not and under what specific conditions it should be done.

A: We agree with your statements about breastfeeding. However, as discussed above, our systematic review did not address breastfeeding but feeding MoM. Furthermore, our study, which is limited to cohorts, provides stronger evidence than previous systematic reviews on perinatal SARS-Cov-2 infection, which included case reports and case series. With the limitations already mentioned, our study supports MoM feeding. With all available data reported, there is no reason not to give MoM, which could help in decision-making.

In order to better respond to your comment, we added to the supplement the results of preliminary results obtained the previous strategy (pages 13-14). This provides insights to guide decisions about whether to breastfeed or not and under what specific conditions it should be done. However, we mention that the certainty about these statements was very low because they only included case series and case reports.

Line 349-351: “Certainty of preliminary results presented in the supplement was very low because these only included case reports and case series. These data did not show any evidence for association of feeding MoM or setting of feeding with neonatal SARS-Cov-2 infection.”

You refer  a previous systematic review (line 74) were you analyzed evidence about delivery management interventions designed to prevent delivery-associated transmission of maternal SARS-CoV-2 to neonates [Ref.19]. Even considering that Breastfeeding and Delivery management are distinct yet interconnected components of neonatal care, especially in the context of preventing viral transmission, the outcome that you analysed depends on both, the discussion or clinical recommendations, integrating them under a unified framework, such as "maternal-neonatal infection prevention strategies", provides a more holistic view. It is understandable that your findings may be limited by the available articles, which often address breastfeeding and delivery management separately. However, there are likely studies that consider both aspects together. Do you consider that this separation might have limited your ability to fully capture potential influences on neonatal infection rates? Could delivery management practices have impacted breastfeeding conditions, thereby affecting your results BB?

A: We initially considered a systematic review assessing unified framework of maternal-neonatal infection prevention strategies to prevent perinatal SARS-CoV-2 transmission. During out initial discussions with the Cochrane Collaboration, it rapidly became evident that the scope would be much too broad; this is why we developed two separate protocols, which initially used the same search and keywords.

Our previous systematic review (which included case reports and case series but no cohorts) showed no evidence to support any association of delivery management with transmission of SARS-Cov-2 virus to the neonate. This was further supported by Boukoura’s systematic review (ref 41), which included cohorts, case reports and case series.

We could not find observational studies with enough granularity to yield a combined analysis. Our repeat searches yielded some cohort studies (e.g., Hudak et al 2023) with information on delivery management and feeding but without the needed granularity to yield a combined strategy.

Manchikanti did not include case reports in ‘Recommendations for systematic reviews of observational studies’ (Manchikanti 2009, PMID: 19787009). Murad et al (Guidelines for systematic reviews of case reports and case series, 10.1136/bmjebm-2017.110853) recommended to use selection as the most important domain of methodological quality of case reports and case series. They suggested using evidence derived from case reports and case series to inform decision making when no higher level of evidence is available.

We submit that merging case series and case reports such as those included in our first systematic review and in Boukoura’s study with the current manuscript - which only includes cohort studies- would decrease, not increase, the certainty of evidence for two reasons. First, comparing dyads with MoM versus without feeding MoM included in the cohorts in the current study allowed us to calculate the risk ratio within each study and to merge all of them into a common risk ratio, which is not possible using case reports or case series. Second, case reports and series, the only publications available initially during the COVID-19 pandemic, selectively included dyads with neonatal SARS-Cov-2 infection. Preliminary data on case series and case reports presented at the Pediatric Academy Society Meeting in 2022 (lines 464-467) suggested a high frequency of neonatal SARS-Cov-2 infection. In these case series and case reports, the frequency associated with any MoM was 33/68 or 50.04 % (95% CI 34.09, 65.98) with any MoM and 17/84 or 30.33% (95% CI 19.21,44.34) with no MoM. These data are presented in the supplementary material, pages 13-14. In contrast, data from the largest cohort presented in the current manuscript (Hudak 2023, ref 25) was only 2.4% with MoM and 1.9% with no MoM.

We added the following statement on lines 97-100: "A detailed history of the search strategy is presented in the supplement. We decided to only include cohorts or randomized trials to prevent selection bias associated with case reports and case series."

Please consider further developing and refining the discussion, conclusion, and future perspectives, including proposed approaches for new studies.

A; We have added a discussion of Boukoura’s systematic review and integrated all literature in the conclusion and future perspectives. (line 381-384): “Another systematic review of breastfeeding with maternal SARS-CoV-2 infection was conducted by Boukara et al. This study supported breastfeeding, skin-to-skin contact and rooming in with diligent hand hygiene, wearing masks and cleansing of breasts only when necessary[41]”.

We changed the conclusion as follows: “Our findings on feeding MoM contribute to the growing body of evidence supporting breastfeeding during maternal SARS-CoV-2 infection.” Lines 440-441.

We thank Reviewer 2 for her/his helpful comments on our manuscript

Round 2

Reviewer 1 Report

Comments and Suggestions for Authors

All comments have adequately addressed.